# Climate Change Influences Basidiome Emergence of Leaf-Cutting Ant Cultivars

**DOI:** 10.3390/jof7110912

**Published:** 2021-10-27

**Authors:** Rodolfo Bizarria, Pepijn W. Kooij, Andre Rodrigues

**Affiliations:** Department of General and Applied Biology, São Paulo State University (UNESP), Rio Claro 13506-900, SP, Brazil

**Keywords:** fungal sexual reproduction, fungiculture, fungus-growing ants, mushroom phenology, nest architecture

## Abstract

Maintaining symbiosis homeostasis is essential for mutualistic partners. Leaf-cutting ants evolved a long-term symbiotic mutualism with fungal cultivars for nourishment while using vertical asexual transmission across generations. Despite the ants’ efforts to suppress fungal sexual reproduction, scattered occurrences of cultivar basidiomes have been reported. Here, we review the literature for basidiome occurrences and associated climate data. We hypothesized that more basidiome events could be expected in scenarios with an increase in temperature and precipitation. Our field observations and climate data analyses indeed suggest that *Acromyrmex coronatus* colonies are prone to basidiome occurrences in warmer and wetter seasons. Even though our study partly depended on historical records, occurrences have increased, correlating with climate change. A nest architecture with low (or even the lack of) insulation might be the cause of this phenomenon. The nature of basidiome occurrences in the *A. coronatus*–fungus mutualism can be useful to elucidate how resilient mutualistic symbioses are in light of climate change scenarios.

## 1. Introduction

Fungiculture evolved in fungus-growing ants about 55–65 million years ago [1,2,3]. To maintain symbiosis homeostasis, ants provide suitable environmental conditions to nurture their fungal partners [4,5,6,7]. Leaf-cutting ants in genera *Acromyrmex* Mayr, 1865; *Amoimyrmex* Cristiano, Cardoso and Sandoval, 2020 [8]; and *Atta* Fabricius, 1804, are notable as dominant herbivores in the Neotropics [7], due to the foraging of a considerable amount of leaves as the substrate for their basidiomycete cultivar (*Agaricales*: *Agaricaceae*: *Leucoagaricus gongylophorus* (Möller) Singer (1986)) [9]. In addition to foraging, leaf-cutting ants are also ecosystem engineers, modifying the environment and performing key ecosystem services [7,10,11,12] such as nutrient cycling [13], seed dispersal and germination [14], and increasing soil fertility and plant growth [15].

The fungal partner of leaf-cutting ants is vertically transmitted as mycelium across generations carried inside the infrabuccal pocket of foundress queens [16,17,18]. Despite the long-term history of clonal transmission, singular field occurrences of basidiomes were reported in this mutualism [19], mostly on shallow nests (Table 1 and Appendix A) [20,21,22]. Asexual transmission performed by leaf-cutting ants contrasts with the preferred sexual reproduction of the fungal partners [19], which may explain the efforts of ants to promptly remove the mushrooms [23,24,25,26], characterizing a conflict of interest between partners over reproduction [19]. Basidiome formation was previously suggested to be regulated by environmental factors in this mutualism, such as temperature and humidity, in laboratory colonies of *Acromyrmex crassispinus* (Forel, 1909) [23], and nest ventilation for field colonies of the leaf-cutting ant *Atta cephalotes* (Linnaeus, 1758) [21].

Symbioses are responsive to biotic and abiotic components of ecosystems [34,35,36,37]. Here, to investigate the influence of climatic parameters in the mushroom phenology of cultivars of leaf-cutting ants, we reviewed environmental conditions for nests with confirmed events of basidiomes in previous reports. Furthermore, we added new reports of recent observations from nests of leaf-cutting ant *Acromyrmex coronatus* (Fabricius, 1804) in Brazil. On the basis of previous mushroom phenology observations [38,39], we hypothesized that, if basidiome occurrences are tied with climate change, more basidiome events are expected with increased temperature and precipitation. Alternatively, basidiome occurrences could be linked with only one of these factors or simply be random events. The nest architecture of *A. coronatus* in tree forks, exposure to warmer and wetter seasons, and the observed fluctuations in temperature and precipitation suggest that climate change increased basidiome occurrences. The nature of these occurrences can be useful to understand how symbiotic systems could be resilient to environmental disturbances, especially in light of climate change scenarios.

## 2. Material and Methods

To establish the current panorama of basidiome occurrences in leaf-cutting ant–fungus mutualism, we reviewed field occurrences since Mueller (2002) [19], adding new reports scattered across the literature and in the Global Biodiversity Information Facility database (GBIF; Table 1 and Appendix A) [40,41]. In addition, basidiomes of *L. gongylophorus* were observed emerging from colonies of *A. coronatus* throughout 2018–2021 in Rio Claro, State of São Paulo, Brazil (Table 1 and Appendix A), and were collected and transferred to the laboratory for mycological evaluation (morphological and molecular analyses, and basidiospore germination assays). Specimens were dried in a food dehydrator, and a representative specimen was deposited at FLOR (Herbarium of the Federal University of Santa Catarina, Department of Botany, Center for Biological Sciences, Florianópolis, Brazil, voucher #FLOR0068416). Basidiospores were collected by spore print from mature basidiomes, i.e., laying the cap directly over glass slides (disinfected with 70% ethanol) and keeping this at room temperature overnight. Collected basidiospores were then picked up using a sterile inoculation loop, suspended in saline solution (0.85% NaCl), followed by serial dilution and inoculation on growth media. Alternatively, basidiospores were collected by fixing a piece of the cap with the gills facing downwards on the lid of Petri plates with an agar fragment, allowing for basidiospores to drop on the culture medium for 24 h at room temperature. In both methods, Potato Dextrose Agar (PDA 3.9%, NEOGENE^®^ Culture Media, Lasing, MI, USA) and Yeast Dextrose Agar (YPD: 1% yeast extract, 2% peptone, 2% glucose, and 1.5% Agar) supplemented with 150 mg mL^−1^ of chloramphenicol (Sigma-Aldrich; St. Louis, MO, USA) were used as culture media, and plates were incubated at 25 °C in darkness.

Fungal genomic DNA was extracted following Lacerda et al. (2018) [42] with modifications. Fungal samples were mechanically broken with glass beads (425–600 µm in diameter; Sigma-Aldrich, St. Louis, MO, USA) in 500 μL sterile lysis buffer (Tris 100 mM, EDTA 10 mM, SDS 2%; pH 8). Then, 5 μL of Proteinase K (20 mg/mL) was added, and samples were incubated at 65 °C for 30 min. Afterwards, 140 μL of 5 M NaCl and 64 μL of 10% CTAB were added followed by incubation at 65 °C for 60 min. Tubes were centrifuged at 10,000 rpm (Eppendorf microcentrifuge, 5424) for 30 s, and 600 μL of chloroform:isoamyl alcohol (24:1) solution was added followed by centrifugation at 12,000 rpm for 10 min. The supernatant was collected and transferred to sterile 1.5 mL tubes. Then, 300 μL of ice-cold absolute isopropanol and 50 μL of 3 M sodium acetate pH 5.2 were added. The suspensions were then centrifuged at 10,000 rpm for 10 min, removed by single inversion, and washed with 600 μL of 70% ethanol. For ethanol removal, the suspensions were centrifuged at 10,000 rpm for 10 min, followed by a single inversion procedure. After ethanol evaporation, 30 μL of Tris EDTA buffer (10 mM Tris; 1 mM EDTA) was added.

For polymerase chain reaction (PCR) amplification, DNA was diluted (1:10 *v*/*v*) in ultrapure sterile water, and PCR reactions were performed in 25 µL reaction volume. The internal transcribed spacer (ITS) region and a fragment of the large subunit (LSU) ribosomal DNA gene were amplified with primer pairs ITS5–ITS4 [43] and LR0R–LR5 [44,45], respectively. PCRs were performed with 4 μL of dNTPs, 5 μL of buffer 5×, 2 μL of 25 mM MgCl_2_, 1 μL of 10 μM forward primer, 1 μL of 10 μM reverse primer, 1 μL of bovine serum albumin (BSA; mg mL^−1^), 8.8 μL of ultrapure sterile water, 0.2 μL of 5 U μL^−1^ Taq polymerase, and 2 μL of diluted DNA (1:10). PCR conditions were as follows: 94 °C/3 min, 35 cycles of 94 °C/1 min, 55 °C/1 min, and 72 °C/2 min. Amplicon purification was performed with Wizard™ SV Gel and PCR Clean-Up System (Promega, Madison, WI, USA), and sequencing reaction was performed using BigDye^®^ Terminator v. 3.1 Cycle Sequencing Kit (Thermo Fisher Scientific, Waltham, MA, USA), according to the manufacturer’s protocols. Sanger sequenced amplicons were injected in ABI 3500 Series Genetic Analyzer (Thermo Fisher Scientific, Waltham, MA, USA). Forward and reverse sequences were edited and assembled in Bioedit v. 7.0.0 [46], and then compared with sequences available in the NCBI-GenBank database with BLASTn.

For each reported basidiome occurrence, we surveyed the respective climate data and information regarding nest architecture, colony health, and ant behavior. Climate data were collected from different databases: Agrometeorological Monitoring System—Agritempo (https://www.agritempo.gov.br/agritempo/index.jsp; data accessed on 13 April 2021), the Meteorological Station at the Center for Environmental Analysis and Planning (CEAPLA; data accessed on 7 April 2020) in Rio Claro, and World Bank Data (http://climateknowledgeportal.worldbank.org/; data accessed on 28 August 2020) for historical data for Rio Claro. To evaluate the influence of climatic fluctuation on basidiome emergence, occurrence counts were compared with the average temperatures (minimal, maximal, and mean) by month and the respective accumulated precipitation (in mm). Analyzes were performed with data for the last six months interval considering the month of basidiome occurrence. Since month interval data did not show homogeneity of variances (Bartlett test, *p* < 0.05; Appendix A), we applied Kruskal–Wallis analysis, followed by Mann–Whitney U tests with Benjamini–Hochberg adjustment of *p* values, both analyses with an alpha threshold of 0.05. Descriptive statistics were performed to explore the effects of climatic variables on the number of basidiomes, by comparing occurrences with anomalies of temperature and precipitation for each period. Partial least squares (PLS) regression coefficients were estimated for each parameter to describe the interaction between predictor (temperature and precipitation) and response variables (number of basidiomes for each occurrence). Variable importance in projection (VIP) scores were estimated to analyze the relative importance of each variable. VIP scores higher than 1 were considered to be the most relevant predictors. Analyses were conducted in RStudio v. 1.4.1717 [47] using R v. 4.1.0 [48] with packages ggplot2 [49] and mdatools [50].

## 3. Results

We recorded a total of 17 occurrences in Rio Claro: 14 new occurrences in 2018–2020 from 6 *A. coronatus* nests (Figure 1A–D), 1 occurrence in 2006 from 1 *A. coronatus* nest, and 2 occurrences in 1996 from 1 *Acromyrmex hispidus fallax* Santschi, 1925 nest (Table 1 and Appendix A). Colonies in which we detected basidiomes remained active and foraging for plant substrate (Table 1 and Appendix A). Ants were cutting pieces of the mature basidiomes after cap opening (Figure 1F) and eventually deformed basidiomes, which were then ignored. We only recorded basidiomes emerging from nests constructed in tree forks by *A. coronatus* (Figure 1A), contrasting with previous observations on shallow or young nests of other leaf-cutting ant species (Table 1). Culturing assays showed that basidiospores of *L. gongylophorus* were viable and able to germinate (sequence accessions: MZ620731-MZ620732 for ITS and MZ618880-MZ618881 for LSU, Appendix A), and the resulting mycelium was able to form gongylidia (i.e., specialized hyphal tip swellings of the fungus cultivated by leaf-cutting ants, Figure 2). Blastn results for these showed over 99% identity, 97% query cover, and an E-value of 0.0 with cultures obtained from the fungus garden of *Acromyrmex coronatus* (nest ID: BLS170701-01; reference accessions: MN473881 and MN473139).

Basidiomes were more frequent during hot and rainy seasons (Figure 3A), from November to April. At the start of these periods, ant workers were carrying pieces of their fungus gardens and pupae (Figure 1E), moving them from underground nests to tree fork nests, with protective behavior during the garden transferring, i.e., ant workers guarding the route against invasions by other ants (Appendix A). PLS regression analysis indicated that (lower) precipitation preceding the month of occurrence of basidiomes can be used to predict the number of basidiomes (fourth month, regression coefficients = 1.3450 ± 0.60, t = 2.19, *p* = 0.044, df = 16; Figure 3B and Appendix A). VIP scores also estimated the importance of precipitation (Appendix A), with the highest scores being 1.82, 1.42, and 1.26 obtained for the fourth, the first, and the second month before basidiome occurrences, respectively. With respect to temperature, the scores mainly indicate the highest influence of two months before the occurrences for the number of observed basidiomes (Appendix A). Analysis of the last six months before occurrence indicates climate similarities between the months closer to the occurrence in contrast to those further away, which registered lower temperatures (Figure 2C–E; Appendix A).

## 4. Discussion

Leaf-cutting ants originated about 19 million years ago [1], establishing a long-term fungal asexual propagation, with limited genetic exchange between cultivar and free-living fungi from natural reservoirs [51]. The potential side effects of long-term clonal transmission [52] and rare occurrences of basidiomes have led to a historical assumption of sexual incompetence of fungal cultivars [19]. However, basidiome occurrences were reported for several leaf-cutting ant species [19] (Table 1 and Appendix A), suggesting that these occurrences may be more frequent than what was previously thought. Here, we recorded several basidiome occurrences on active *A. coronatus* nests constructed in tree forks. Occurrences were frequent during hot and rainy seasons, mainly after dry seasons.

The sexual reproduction of fungal cultivars is a considerable investment that contrasts with the efforts performed by ants on fungal clonal transmission [19], mainly with the production and release of viable basidiospores. This assumption was supported by observations of attempts by leaf-cutting ants to rapidly remove fungal sexual structures [23,24,25,26], suppressing the primordial basidiomes or even cutting away the gills with basidiospores (Figure 1F). Since the pioneering study of Möller (1893), basidiomes have not been confirmed to produce viable basidiospores, as shown by repeated failed attempts of basidiospore germination [23,25,26]. Here, a considerable amount of basidiospores were sampled from basidiomes emerging from *A. coronatus* colonies and culturing attempts confirmed viability, supporting the previous observations by Möller (1893) [20]. The implications of fungal basidiospore viability and dispersion probability, and the differences in cultivar growth vigor (i.e., morphologies) from obtained cultures should be further investigated.

Previous occurrences of basidiomes in the leaf-cutting ant–fungus mutualism were recorded from shallow nests [20,22] or from colonies of ant species that build shallow nests (Table 1 and Appendix A). In contrast to deep underground nests, shallow nests are more exposed to climatic fluctuations due to a lack of insulation of soil layers or plant material (leaf fragments, twigs, etc.), which may explain why most of the field occurrences were recorded from *Acromyrmex* colonies, since this genus generally creates more shallow nests [19]. *Acromyrmex coronatus* shows particular plasticity of nest architecture [28], building deep or shallow belowground nests and nests in tree forks. Basidiomes were only observed on *A. coronatus* nests in trees, which are more likely to be exposed to climatic fluctuations and nest ventilation than belowground nests that are able to maintain a more stable environment [17]. *Acromyrmex coronatus* workers were also observed moving and carrying pieces of fungus garden at the beginning of the rain seasons (Figure 1E, Appendix A) despite the risk of garden contamination and drying, which would eventually lead to garden loss. These findings, together with the basidiome occurrences, lead to a seasonal influence on the *A. coronatus* mutualism, mainly through warmer and wetter seasons, and in particular by increasing precipitation in the months before the occurrences.

Climatic effects on fungal basidiomes were also reported in other biological systems, such as in the mutualism between fungus-growing termites and *Termitomyces* fungi [53], mycorrhizal fungi (e.g., *Tricholoma matsutake* and *Tuber* spp.) [38,39,54], and epigeous fungi occurring in forest ecosystems [55,56]. The seasonal influence on the *A. coronatus*–fungus mutualism might increase the basidiome occurrences, which supports our hypothesis of more basidiome events with increased temperature and precipitation. However, further research is necessary to fully reject the alternative hypothesis, which considers a stochastic nature for these events. The influence of climate change on basidiome occurrences might be intensified by the nest architecture and should be further investigated for *A. coronatus,* similar to research for *A. ambiguus* and *A. heyeri* [4,5]. Although climatic data here were obtained from different meteorological field stations and not simultaneously measured across all sampling locations, the general basidiome occurrence profiles were similar, and observed in warmer and wetter seasons. The influence of specific parameters that modulate the occurrence and the number of basidiomes should also be further investigated, to avoid data bias. Similarly, the reasons behind the abundance of recent field observations in contrast with the rarity of such reports should be carefully investigated in light of climate disturbances. This scenario advances our knowledge on fungus-growing ants ecology and provides a symbiotic model system to study responses to climate disturbances, which are expected to impact biodiversity and ecosystems [57,58].

## 5. Conclusions

Using field observations of *A. coronatus* colonies combined with climate data analyses, we showed that basidiome occurrences in colonies of this leaf-cutting ant increase during the warmer and wetter seasons. The nest architecture of *A. coronatus* colonies with likely low insulation might be the cause of this phenomenon. Furthermore, these events are intensified considering the current climate change scenarios. Although our study partly relies on historical records of basidiomes in the attine ant mutualism, our observations contrast with the rarity of this phenomenon, and further investigation is necessary in light of climate change and the stability of this mutualism.

## Figures and Tables

**Figure 1 jof-07-00912-f001:**
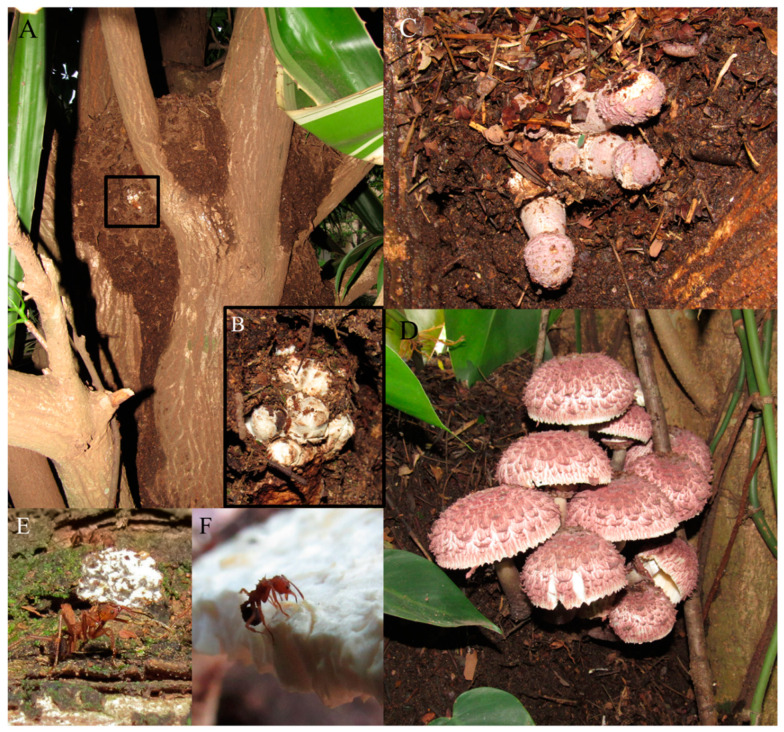
Basidiome occurrence in *Acromyrmex coronatus* nest in Rio Claro, State of São Paulo, Brazil. (**A**,**B**) Basidiome primordia in a nest located in a tree fork. (**C**,**D**) Emerged basidiomes. (**E**) Garden transfer by an *A. coronatus* worker (see also Appendix A for video). (**F**) *A. coronatus* worker attempting to cut parts of the basidiome.

**Figure 2 jof-07-00912-f002:**
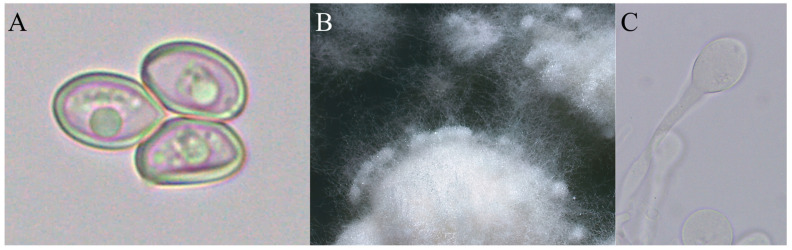
*Leucoagaricus gongylophorus* cultured from basidiospores. (**A**) Basidiospores obtained from a basidiome collected on 21 February 2020 in Rio Claro-SP, Brazil (Colony ID: RB200507-03). (**B**) Fungal mycelium on potato dextrose agar medium after incubating for 30 days at 25 °C showing typical growth pattern for *L. gongylophorus* with staphylae (clusters of gongylidia) at the edge. (**C**) Gongylidia (i.e., specialized hyphal tip swellings) found in the cultured mycelium.

**Figure 3 jof-07-00912-f003:**
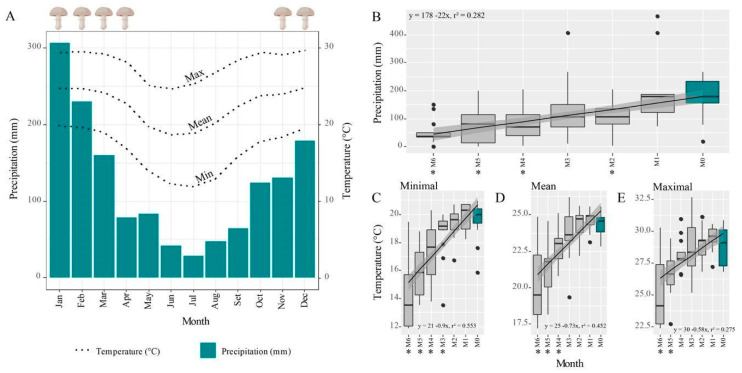
Basidiomes were more frequent during warm and wet seasons. (**A**) Annual climate panorama (1994–2021) of Rio Claro (State of São Paulo, Brazil) with basidiome occurrences from November to April. (**B**–**E**) Six-month interval of climatic data between occurrences (in blue) and months preceding them. Overall, no differences were found between the two months preceding the occurrences (see details of analysis in Appendix A), which followed the periods with increased temperature and precipitation. Asterisks indicate statistical differences for the Mann–Whitney U test between the previous months and the month of basidiome occurrence.

**Table 1 jof-07-00912-t001:** Field occurrences of basidiomes in leaf-cutting ant nests. Data reviewed in the literature were compiled with new occurrence reports since the publication by Mueller (2002) [19]. A full version of the table is available in Appendix A.

Ant Species	Fungal Species	Occurrence Data	References
*Acromyrmex aspersus*	*Leucoagaricus gongylophorus*	Basidiomes found on 29 November 1995 in an active nest in a native forest in Santa Cruz do Sul-RS (Rio Pardinho), Brazil.	[27]
*Acromyrmex coronatus*	*Leucoagaricus gongylophorus*	Basidiomes observed on 23 March 2006, in Rio Claro-SP (UNESP Campus), Brazil.	Bacci M. (personal communication)
*Acromyrmex coronatus*	*Leucoagaricus gongylophorus*	Basidiomes observed on 5 and 27 January 2018 in Rio Claro-SP, Brazil (Colony ID: BLS170701-01).	This study
*Acromyrmex coronatus*	*Leucoagaricus gongylophorus*	Basidiomes observed on 5 April and 27 November 2018, 28 January 2019, 10 February 2020, in Rio Claro-SP, Brazil (Colony ID: RB181203-01). One specimen is deposited in the herbarium of the Federal University of Santa Catarina # FLOR0068416).	This study
*Acromyrmex coronatus*	*Leucoagaricus gongylophorus*	Basidiome observed on 29 December 2019 in Mogi-Guaçu-SP, Brazil (colony ID: RB200104-01).	This study
*Acromyrmex coronatus*	*Leucoagaricus gongylophorus*	Basidiomes observed on 2 and 10 January 2020, 5 and 7 February, 12 November, and 13 and 18 December 2020 in Rio Claro-SP, Brazil (Colony ID: RB190909-01).	This study
*Acromyrmex coronatus*	*Leucoagaricus gongylophorus*	Basidiome observed on 4 January 2020 in Mogi-Guaçu-SP, Brazil (Colony ID: RB200104-03).	This study
*Acromyrmex coronatus*	*Leucoagaricus gongylophorus*	Basidiome observed on 21 February 2020, in Rio Claro-SP, Brazil (Colony ID: RB200507-03).	This study
*Acromyrmex crassispinus*	*Rozites gongylophora* (=*Leucoagaricus gongylophorus*)	As mentioned in Gonçalves (1961, page 117) “Luederwaldt 1926 observed basidiomes on nests of *A. crassispinus* (cited as *A. nigra*).	[19,28,29]
*Acromyrmex disciger*	*Leucocoprinus gongylophorus* (=*Leucoagaricus gongylophorus*)	Basidiomes observed in November 1891, 19 February, and 17 and 30 March 1892 in Blumenau-SC, Brazil.	[19,20,30]
*Acromyrmex hispidus fallax*	*Rozites gongylophora* (=*Leucoagaricus gongylophorus*)	Observation from 11 November 1944 in Curitiba-PR, Brazil (see page 143).	[19,28]
*Acromyrmex hispidus fallax*	*Leucoagaricus gongylophorus*	Last week of February 1996 (dozen of basidiomes) and April (ten basidiomes) in Rio Claro-SP, Brazil. Colony found under a peach tree.	[19,22]
*Atta cephalotes*	Unidentified	No data/not sure	[19,31]
*Atta cephalotes*	Unidentified	Lelydorp, 20 km near Paramaribo, Suriname.	[19,32,33]
*Atta cephalotes*	*Rozites gongylophora* (=*Leucoagaricus gongylophorus*)	8 and 9 November 1939 in Paramaribo, Suriname (see pages 253–254, and 258–266).	[19,21]
*Atta cephalotes*	*Rozites gongylophora* (=*Leucoagaricus gongylophorus*)	13 January 1940, in Paramaribo, Suriname (see page 260).	[19,21]
*Atta colombica*	*Leucocoprinus* cf. *gongylophorus*	No data/not sure	Collection PA-236, U.G. Mueller, unpublished data
Not informed	*Rozites gongylophora*	Basidiome observed on 29 May 1957.	gbif.org/occurrence/2464956731, last accessed on 25 June 2021
We assumed *Acromyrmex* sp. on the basis of the collector’s ant-colony descriptions	*Leucoagaricus gongylophorus*	Basidiome observed on 22 December 2010 in Brasília-DF, Brazil. Sh Jardim Botânico/Condomínio Quintas Bela Vista Conjunto e—Jardim Botânico, Brasília—DF, 70297-400, Brasil (Coordinates: 15.9 S and 47.8 W).	gbif.org/occurrence/1986496332, last accessed on 25 June 2021
We assumed *Acromyrmex* sp. on the basis of collector’s ant-colony descriptions	*Leucoagaricus gongylophorus*	Basidiome observed on 11 November 2017 in Parque Nacional da Serra dos Órgãos—Parnaso, Teresópolis-RJ, Brazil.	Heisecke C., Duque J. and Venegas M. (personal communication)

## Data Availability

Data supporting the results in the paper are available in the Appendix A.

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
