# Peer review of "Climate Change Influences Basidiome Emergence of Leaf-Cutting Ant Cultivars"

_jof, 2021, doi:10.3390/jof7110912_

Round 1

Reviewer 1 Report

Comment 1: Really appreciate authors field work, but looks like just part of materials and methods. Need more detailed investigation.

Comment 2: Please include materials and methods of supplementary information in main text. If possible include data and figures about molecular biological results.

Comment 3: Include agar plates results, if possible microscopic results of your investigation.

Comment 4: Please improve materials and methods and results section with more scientific information.

Comment 5: Write clear conlcusion

Reviewer 2 Report

The paper suggests that basidiome formation, as hypothesized, is due to temperature, humidity, and nest ventilation. The photos are good, and the paper is concise and well structured. Other than quite a few English errors, the paper is fine. Though some may have concerns with the low sample size, I know this is a rare phenomenon to observe.

Reviewer 3 Report

Data presented suggests that this is the preliminary study. Please describe it in the discussion. Please add limitations of the study.

Reviewer 4 Report

    The manuscript is well written and in a coherent manner. The stated conclusions correlate to the obtained results. The article should be interesting for specialists, but less interesting for researchers specialized in other fields or for other types of readers. 

Round 2

Reviewer 1 Report

Comment 1: Please improve abstract.

Comment 2: If possible, please include both Fig S1 and S2 as main figures.

Comment 3: Please write separate section for Conclusion
